# A Graph Transformer Framework for Multi-step Prediction of Time-domain Maxwell's Equations

## Abstract

Spatiotemporal modeling of electromagnetic fields governed by time-domain Maxwell's equations is essential for simulating and understanding wave propagation and scattering phenomena. However, accurate long-term predictions remains challenging due to the stringent numerical stability requirements and high computational costs inherent in traditional numerical algorithms. We propose **GT-MSMW**, a specialized framework built upon the finite-difference time-domain (FDTD) method, which integrates graph neural networks (GNNs) with a residual Transformer to enable efficient and accurate multi-step forecasting of time-domain Maxwell's equation solutions for the first time. Unlike previous neural methods that rely on step-by-step autoregressive propagation, **GT-MSMW** directly maps the initial field distribution to the desired state, thus mitigating cumulative errors. To ensure both accuracy and flexibility, the proposed model uses unstructured mesh discretization, GNNs to capture dominant spatial interactions, and the Transformer to model remaining long-range dependencies. Extensive experiments across various 2D and 3D electromagnetic scattering scenarios demonstrate that **GT-MSMW** achieves superior accuracy and generalization, offering a powerful data-driven solver for Maxwell-based simulations.

## 1 Introduction

The finite-difference time-domain (FDTD) method is a classic numerical approach widely used for electromagnetic simulations (Sullivan, 2013). It discretizes space and time to convert Maxwell's equations into a set of difference equations for efficient computation. FDTD excels at modeling broadband signal propagation, making it essential for applications in metamaterial modeling (Hao & Mittra, 2008), nanophotonics (Gallinet et al., 2015), photonic device design (Zeng et al., 2021), and antenna analysis (Jensen & Rahmat-Samii, 2002). However, traditional FDTD is limited by the Courant-Friedrichs-Lewy (CFL) condition (Taflove & Hagness, 2005), requiring very small time steps to ensure stability in large-scale simulations. This leads to a longer computation time for long-term evolution, making it difficult to quickly obtain stable results. Additionally, due to grid meshing and stability issues, simulating complex or heterogeneous materials, such as biological tissues, will further increase computational costs and reduce the efficiency in practical applications.

The rapid advancement of artificial intelligence has revitalized the traditional FDTD method. By integrating deep neural networks with FDTD, more efficient time-domain simulation algorithms have emerged, significantly enhancing both the computational speed and accuracy of electromagnetic simulations (Desai et al., 2022; Yao & Jiang, 2019; Li et al., 2020; Chen et al., 2024). Moreover, the differentiable nature of neural networks enables inverse design, allowing material parameters and geometric structures to be optimized directly from target performance (Mahlau et al., 2025). This has greatly accelerated the development of complex photonic and microwave devices.

Deep learning-based FDTD methods can be broadly categorized into two approaches: global surrogate modeling and single-step surrogate modeling. The global surrogate modeling approach establishes a direct mapping between structural parameters and simulation outcomes, bypassing the iterative computations of traditional FDTD to achieve significantly faster simulations. Sullivan et al. developed a surrogate model for simulating the optical properties of microstructured materials,

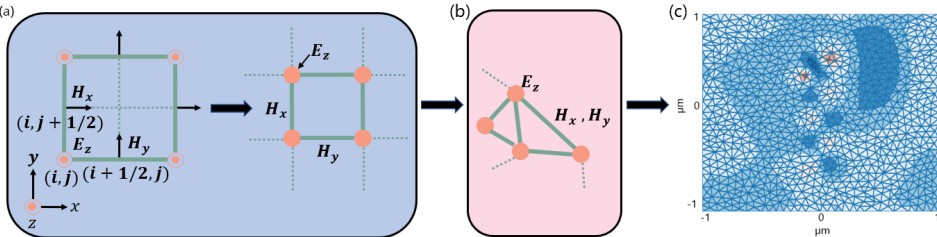

Figure 1: The process of constructing the graph structure. (a) Left: Yee grid for 2D TM polarization. Right: Graph representation on a regular grid. (b) Extension to an unstructured, non-uniform triangular mesh, with $E_z$ embedded in node features and $H_x$, $H_y$ encoded in edge features. (c) An example of a 2D triangular mesh.

greatly improving efficiency and enabling microstructure design and optimization through inverse neural networks (Sullivan et al., 2023). This approach has also been successfully applied to the simulation and design of all-optical plasmonic switches (Adibnia et al., 2024), markedly enhancing the efficiency of forward modeling and inverse design for optical devices. However, global surrogate models suffer from limited generalization, performing well only in specific scenarios and struggling with unseen problems.

In contrast, the single-step surrogate modeling approach retains the iterative electromagnetic field updates of traditional FDTD, leveraging deep learning to accelerate specific computations within each iteration, offering superior generalization. A notable example is the FDTD-RCNN method (Guo et al., 2023), which embeds finite-difference operators into convolution kernels and uses recurrent neural network to simulate time stepping, achieving results consistent with traditional FDTD solvers without training. Mahlau et al. advanced this further by implementing FDTD simulations of 3D photonic nanostructures directly on GPU platforms (Mahlau et al., 2025), utilizing the automatic differentiation capabilities of deep learning frameworks to enable inverse design of nanostructures. However, the performance gains of such methods largely depend on GPU computational power, making them less effective for long-term predictions in large-scale or complex structures.

To better utilize neural networks, Li et al. proposed a multilayer perceptrons (MLPs)-based FDTD method (Li et al., 2020), replacing field update equations with MLP networks to significantly reduce time complexity. Similarly, Kuhn et al. introduced graph neural networks (GNNs) to adapt to varying computation domains and diverse object geometries, enabling efficient iterative solutions to Maxwell's equations (Kuhn et al., 2023). However, their findings highlight that iterative multi-step predictions, where network outputs are fed back as inputs, suffer from accumulated prediction errors, limiting their effectiveness. To address this, Noakoasteen et al. explored a Transformer-based approach (Noakoasteen et al., 2024), predicting field distributions for the next five time steps using 5 or 10 prior frames. This method achieved a 14-fold speed increase over traditional FDTD solver, demonstrating the feasibility of multi-step predictions.

The combination of deep learning and the FDTD method has shown remarkable potential in improving computational efficiency and enhancing design optimization. However, most existing approaches rely on the FDTD paradigm, predicting future states based on one or more previous steps (Kuhn et al., 2023; Noakoasteen et al., 2024) , which poses challenges for generalization and long-term prediction accuracy. Therefore, we propose a multi-step prediction framework that, to the best of our knowledge, is the first to enable direct, end-to-end forecasting of the field at an arbitrary time step $t = n$ from the initial state $t = 0$. The model architecture is primarily built upon GNNs, with Transformer blocks appended as residual components. We refer to this framework as the **G**raph **T**ransformer for **M**ulti-**s**tep Prediction of Time-domain **Max**well's Equations (**GT-MSMW**). Our key contributions are as follows:

- We qualitatively explain the adaptability of **GT-MSMW** in multi-step prediction of time-domain Maxwell's equation based on FDTD paradigm. Specifically, for a given node, the spatial region of its influence from the initial state increases with the time step $n$. While the most significant influence typically originates from neighbors and is effectively captured by the GNN, the remaining long-range dependencies are modeled by the Transformer module.

- Our experiments consider both two-dimensional (2D) transverse magnetic (TM) polarization and three-dimensional (3D) electromagnetic wave propagation scenarios, each involving 100 distinct scatterers, with the objective of predicting the evolution of the electromagnetic field over 100 time steps. To assess the model's generalization ability, we evaluate its performance under varying spatial resolutions and excitation frequencies. Comparative experiments demonstrate the robustness and superior performance of the proposed model. Furthermore, ablation studies validate the key insight that spatial graph priors contribute more significantly than global attention mechanisms in solving time-domain Maxwell's equations, thereby supporting the rationale behind our model architecture.

## 2 PRELIMINARIES

### 2.1 TIME-DOMAIN MAXWELL'S EQUATIONS

In this part, we start from the source-free time-domain Maxwell's equations under 2D TM polarization, where the electric field $E_z$ is normal to the $x$-$y$ plane, while the magnetic fields $H_x$ and $H_y$ are confined within it. The 3D scenario, which involves additional distinctions, is discussed in Appendix B.2.

$$\frac{\partial E_z(\boldsymbol{x}, t)}{\partial y} = -\sigma_m H_x(\boldsymbol{x}, t) - \mu \frac{\partial H_x(\boldsymbol{x}, t)}{\partial t}, \tag{1}$$

$$\frac{\partial E_z(\boldsymbol{x}, t)}{\partial x} = \sigma_m H_y(\boldsymbol{x}, t) + \mu \frac{\partial H_y(\boldsymbol{x}, t)}{\partial t}, \tag{2}$$

$$\frac{\partial H_y(\boldsymbol{x}, t)}{\partial x} - \frac{\partial H_x(\boldsymbol{x}, t)}{\partial y} = \sigma E_z(\boldsymbol{x}, t) + \varepsilon \frac{\partial E_z(\boldsymbol{x}, t)}{\partial t}. \tag{3}$$

Here, $\boldsymbol{x} = (x, y)$; $\sigma(\boldsymbol{x})$ and $\sigma_m(\boldsymbol{x})$ represent the electric and magnetic conductivities; and $\varepsilon(\boldsymbol{x})$ and $\mu(\boldsymbol{x})$ correspond to the spatially varying permittivity and permeability.

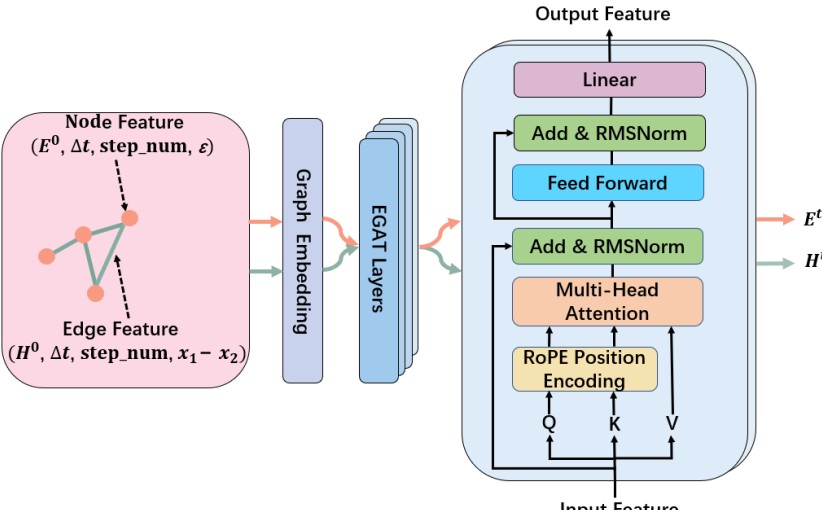

Figure 2: The model architecture of **GT-MSMW** based on FDTD.

### 2.2 GRAPH TRANSFORMER

**GNNs.** A graph $\mathcal{G}(\mathcal{V}, \mathcal{E})$ consists of a node set $\mathcal{V} = \{v_1, v_2, \dots\}$ and an edge set $\mathcal{E} \subseteq \mathcal{V} \times \mathcal{V}$. Each directed edge $e_{ij} = (v_i, v_j) \in \mathcal{E}$ represents a connection from node $v_i$ to node $v_j$, while in undirected graphs, edges imply bidirectional connectivity.

The core idea of GNNs is to update node features by aggregating information from neighboring nodes and edges, allowing each node to encode both its local features and structural position within the graph (Gilmer et al., 2017; Scarselli et al., 2009; Pearce et al., 2021; Dwivedi et al., 2022;

Hamilton et al., 2017). Graph convolutional networks (GCN) (Kipf & Welling, 2017) achieve this by applying spectral-based convolutions, enabling efficient feature propagation. Graph attention networks (GAT) (Veličković et al., 2018) further enhance this process by incorporating attention mechanisms, dynamically adjusting the importance of different neighbors to improve robustness and interpretability.

Edge-featured graph attention networks (EGAT) (Wang et al., 2021) extend GAT by integrating edge features into the attention mechanism, enabling simultaneous updates of node and edge representations for more expressive graph modeling. The update process is detailed in Eq. 4.

$$
\begin{aligned}
f'_{ij} &= \text{LeakyReLU}(A[h_i||f_{ij}||h_j]), \\
e_{ij} &= \vec{F}(f'_{ij}), \\
\alpha_{ij} &= \text{softmax}(e_{ij}), \\
h'_i &= \sum_{j \in N_i \bigcup \{i\}} \alpha_{ij} W_t h_j.
\end{aligned}
\tag{4}
$$

where $f'_{ij}$, $h'_i$ are updated node and edge features respectively, $A \in \mathrm{R}^{N \times N}$ represents the weight matrix, LeakyReLU is the leaky rectified linear unit activation function, $\vec{F}$ is the weight vector and $e_{ij}$ denotes unnormalized attention scores.

**Transformer.** Since its introduction, the Transformer architecture has achieved remarkable success across various domains (Latif et al., 2023; Kalyan et al., 2022). It is primarily based on a self-attention mechanism that enables the model to capture long-range dependencies in input sequences. In addition, the multi-head attention mechanism allows the model to implicitly learn representations from multiple perspectives.

**Related Works.** GNNs have shown promising results in spatiotemporal partial differential equations (PDEs) on unstructured, non-uniform meshes (Shen et al., 2025; Zeng et al., 2025). In contrast to traditional neural networks, such as multilayer perceptrons (MLPs) used in physics-informed neural networks (PINNs), which require fixed input shapes (Raissi et al., 2019; Kharazmi et al., 2021), and convolutional neural networks (CNNs), which rely on fixed-scale convolutional operations to extract local features from regular grids (Özbay et al., 2021), GNNs utilize message passing on graphs that naturally accommodate irregular and arbitrarily sized mesh geometries. This flexibility enables robust generalization across meshes with varying resolutions and connectivity patterns, effectively addressing a key challenge in model transferability. Consequently, GNNs have been successfully applied across diverse fields, including materials science (Shi et al., 2024), chemistry (Reiser et al., 2022), and fluid dynamics (de Avila Belbute-Peres et al., 2020; Li & Farimani, 2022).

However, GNNs also exhibit several limitations, such as over-smoothing (Rusch et al., 2023; Scholkemper et al., 2024), over-squashing (Topping et al., 2022), and challenge in capturing long-range dependencies (Dai et al., 2018). Therefore, Graph Transformers (GTs) (Hoang & Lee, 2024; Kreuzer et al., 2021; Müller et al., 2024) have recently emerged as competitive alternatives to GNNs, addressing inherent limitations of neighborhood aggregation paradigms. By employing global self-attention mechanisms (Vaswani et al., 2023), GTs enable direct interactions between any pair of nodes, irrespective of their adjacency or proximity. Regarding the integration method, Rampášek et al. (Rampášek et al., 2023) emphasize well-structured positional and structural encodings, proposing the general, powerful, scalable (GPS) Graph Transformer with linear complexity and state-of-the-art performance. In contrast, Min et al. (Min et al., 2022) explore Graph-Transformer interactions, highlighting three key design aspects: utilizing GNNs as auxiliary modules, improving positional embedding from graphs, refining attention matrix from graphs.

## 3 METHODOLOGY

### 3.1 FDTD: THE NUMERICAL BASIS OF OUR FRAMEWORK

The FDTD method uses the Yee grid (Yee, 1966), which employs staggered electric and magnetic field components for spatial and temporal discretization. This compact setup enhances computational accuracy and efficiency. The computational domain is discretized with indices $i$ and $j$ corresponding to the $x$- and $y$-axes, respectively, with positions $\boldsymbol{x} = (x, y) = (i\Delta x, j\Delta y)$, and temporal

steps $t_n = n\Delta t$. For notational convenience, we henceforth denote $\boldsymbol{x} = (i,j)$ and $t = n$. The electric field $E_z$ and magnetic fields $H_x, H_y$ are spatially offset by $\Delta x/2$, $\Delta y/2$, and staggered temporally by $\Delta t/2$, enabling accurate finite-difference approximations. The update equations for the fields are as in Eqs. 5-7:

$$H_x^{n+\frac{1}{2}}(i,j+\tfrac{1}{2}) = CP(m) \cdot H_x^{n-\frac{1}{2}}(i,j+\tfrac{1}{2}) - CQ(m) \cdot \frac{E_z^n(i,j+1)-E_z^n(i,j)}{\Delta y} \tag{5}$$

$$:= f_{H_x}\big(H_x^{n-\frac{1}{2}}(i,j+\tfrac{1}{2}), E_z^n(i,j+1), E_z^n(i,j)\big),$$

$$H_y^{n+\frac{1}{2}}(i+\tfrac{1}{2},j) = CP(m) \cdot H_y^{n-\frac{1}{2}}(i+\tfrac{1}{2},j) + CQ(m) \cdot \frac{E_z^n(i+1,j)-E_z^n(i,j)}{\Delta x} \tag{6}$$

$$:= f_{H_y}\big(H_x^{n-\frac{1}{2}}(i+\tfrac{1}{2},j), E_z^n(i+1,j), E_z^n(i,j)\big),$$

$$E_z^{n+1}(i,j) = CA(m) \cdot E_z^n(i,j) + CB(m) \cdot \Big[ \frac{H_y^{n+\frac{1}{2}}(i+\frac{1}{2},j)-H_y^{n+\frac{1}{2}}(i-\frac{1}{2},j)}{\Delta x} \tag{7}$$

$$- \frac{H_x^{n+\frac{1}{2}}(i,j+\frac{1}{2})-H_x^{n+\frac{1}{2}}(i,j-\frac{1}{2})}{\Delta y}\Big]$$

$$:= f_{E_z}\big(E_z^n(i,j), H_y^{n+\frac{1}{2}}(i+\tfrac{1}{2},j), \cdots, H_x^{n+\frac{1}{2}}(i,j-\tfrac{1}{2})\big).$$

where the corresponding coefficients $CA(m), CB(m), CP(m), CQ(m)$ are presented in Appendix B.1, where the index $m$ takes values corresponding to the spatial locations of the field components on the left-hand side of Eqs. 5-7. Specifically, we define $f_{H_x}$, $f_{H_y}$, and $f_{E_z}$ to represent the update expressions for $H_x$, $H_y$, and $E_z$, respectively.

To analyze how the initial field at time $t = 0$ influences the field at time $t = n$, we recursively expand the update equations forward in time. For illustration, we take $E_z^n(i,j)$ as a representative example in 8, where $\circ$ denotes function composition and $* \in \{H_x, H_y, E_z\}$.

$$E_z^n(i,j) = f_{E_z}\big(E_z^{n-1}(i,j), \cdots, H_x^{n-\frac{1}{2}}(i,j-\tfrac{1}{2})\big)$$

$$= f_{E_z}\big(f_{E_z}(E_z^{n-2}(i,j), \cdots, H_x^{n-\frac{3}{2}}(i,j-\tfrac{1}{2})), \cdots,$$

$$f_{H_x}(H_x^{n-\frac{3}{2}}(i,j+\tfrac{1}{2}), E_z^{n-1}(i,j+1), E_z^{n-1}(i,j))\big) \tag{8}$$

$$= f_{E_z} \circ \cdots \circ (f_{E_z}(E_z^0(i,j), \cdots), \cdots, f_{H_x}(H_x^{\frac{1}{2}}(i,j-\tfrac{1}{2}), \cdots)).$$

$$= f_{E_z} \circ \cdots \circ f_*(x \in \mathcal{N}_n(i,j))$$

Through Eq. 8, we obtain the effective receptive region at time step $n$ and spatial location $(i,j)$, denoted as $\mathcal{N}_n(i,j)$, which refers to the set of spatial locations at the initial time step whose values contribute to the computation of $E_z^n(i,j)$. As the time step $n$ increases, the number of recursive update operations grows, thereby enlarging the spatial region influenced by the initial field. Consequently, the effective receptive region $\mathcal{N}_n(i,j)$ expands with $n$, indicating that increasingly distant initial values $E_z^0$, $H_x^0$, and $H_y^0$ begin to affect the evolution of $E_z^n(i,j)$ over time. The same conclusion holds for $H_x^n(i,j)$ and $H_y^n(i,j)$. Strictly speaking, $H_x^n$ and $H_y^n$ refer to the field values at time $t = (n + \frac{1}{2})\Delta t$; however, for notational convenience, we also denote them as step $n$.

Another key point is that, for a given node, the influence from its nearby neighbors is typically stronger, while the influence from distant nodes weakens as the distance increases. Accordingly, model architectures should primarily emphasize local information, with distant region integrated as supplementary context.

**Adaptability of GT-MSMW.** The message-passing mechanism of GNNs, which aligns naturally with the local update rules in 8, effectively extracts local node features. However, as the number of time steps $n$ increases, the spatial region $\mathcal{N}_n(i,j)$ influencing the target node $(i,j)$ expands, and GNNs become limited in capturing such long-range dependencies. To address this limitation, **GT-MSMW** combines GNNs to model dominant local interactions with Transformers to capture the broader, long-range dependencies. This hybrid design alleviates the over-smoothing issue caused by simply stacking more GNN layers.

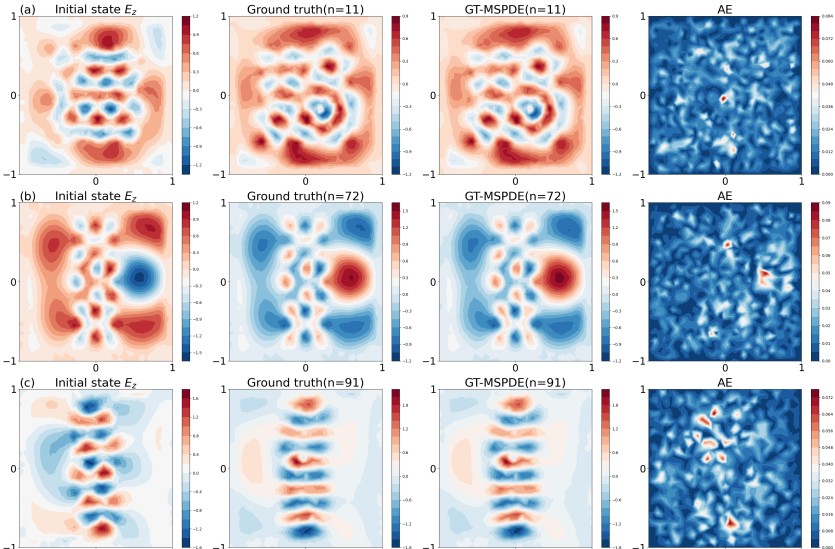

Figure 3: Comparisons of $E_z$ between the ground truth and **GT-MSMW**. (a)–(c) represent three different examples from the test set of **2D-R1F0**. The first column shows the initial states of $E_z$; the second column shows the ground truth; the third column displays the results predicted by **GT-MSMW**; and the last column presents the corresponding AE distribution.

### 3.2 GRAPH STRUCTURE

Owing to the coupled nature of the Yee grid in FDTD, we construct the graph structure shown in Fig. 1. FDTD typically produce regular grids, which may lack the flexibility and precision necessary to accurately represent irregular domains, potentially resulting in computational artifacts. To overcome this limitation, **GT-MSMW** generalizes regular grids to an extended unstructured, non-uniform triangular mesh, similar to the approach in Kuhn et al. (2023).

To enable the simultaneous update of both node and edge features, we adopt EGAT as the core GNN architecture. The graph encodes the initial electric field $E_z^0$ into the input node features, while the initial magnetic field components $H_x^0$ and $H_y^0$ are embedded into the input edge features. To facilitate direct prediction of field values at the $n$-th time step from the initial state, we explicitly incorporate temporal information, including the time step size $\Delta t$ and the step index $\mathrm{step\_num}(n)$ into both node and edge features. In addition, for scenarios involving dielectric materials, the spatially varying permittivity $\epsilon(\boldsymbol{x})$ is included in the node features. Structural information is also embedded by incorporating the relative spatial displacement $\boldsymbol{x}_d = \boldsymbol{x}_1 - \boldsymbol{x}_2 \in \mathbb{R}^2$ between connected nodes into the edge features.

**GT-MSMW** begins with two separate encoders that project node and edge features into a high-dimensional latent space. Each encoder consists of four fully connected layers with ReLU activations. Following the encoding stage, four EGAT layers, each equipped with two attention heads, are stacked to enable iterative feature propagation and interaction.

### 3.3 TRANSFORMER STRUCTURE

The self-attention mechanism in Transformer the input tokens as a fully connected graph, which helps alleviate the limited receptive field of traditional GNNs. This allows the model to emphasize local information while also capturing long-range dependencies from distant field values.

Therefore, we adopt a two-layer encoder-only Transformer architecture for subsequent processing of the input features, with each layer utilizing 8 attention heads. Given the fundamental role of spatial topology in FDTD, we argue that the relative positioning of nodes is crucial. To better capture such relationships, we employ rotary position embeddings (RoPE) (Su et al., 2023) instead of traditional positional encodings, which have been shown to be less effective in preserving relative positional information after linear transformations (Le QI, 2023).

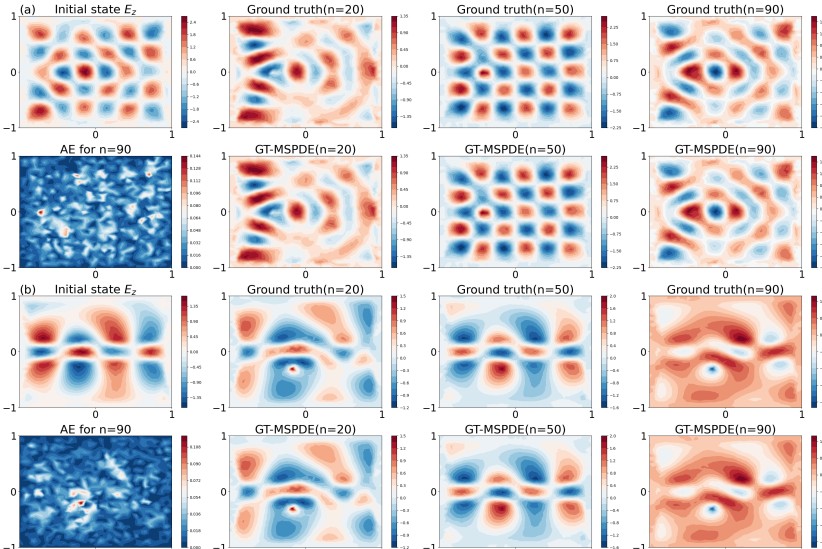

Figure 4: Visualization of $E_z$ predicted by **GT-MSMW** at n=20, n=50, and t=90 for two examples from the **2D-R1F0** dataset, each with distinct initial conditions. In particular, the AE at $n = 90$ is also visualized, as long-term predictions are generally more challenging. This demonstrates the high accuracy and stability of **GT-MSMW** at later time steps.

In addition, to reduce the computational cost associated with the fully connected attention graph, we utilize two lightweight Transformer modules to separately generate the updated $E_t$ and $H_t$. This design does not compromise the interaction between electric and magnetic fields, as such coupling is already handled within the GNN module. The overall model framework is illustrated in Fig. 2, and the corresponding pseudocode is provided in Appendix A.

Table 1: Summary of dataset configurations

| Dataset | Resolution $pixels/\mu m$ | Frequency $\mu m^{-1}$ | Samples | Time Steps |
|---|---|---|---|---|
| **2D-R0F0** | 60 | 1 | 100 | 100 |
| **2D-R1F0** | [40, 80] | 1 | 100 | 100 |
| **2D-R1F1** | [40, 80] | [1, 2] | 100 | 100 |
| **3D-R0F0** | 60 | 1 | 100 | 100 |

## 4 NUMERICAL EXPERIMENTS

### 4.1 EXPERIMENTAL SETUP

**Data.** We construct the datasets using the open-source FDTD solver Meep (Oskooi et al., 2010) and the Python library MeshPy (Steinbrecher & Popp, 2021) to generate triangular meshes. It is worth noting that, although the FDTD method is inherently defined on a regular grid, Meep provides approximate field values at arbitrary spatial locations and time steps. This capability allows us to effectively generate both input and ground truth data on unstructured, non-uniform triangular meshes.

In FDTD simulations, the choice of spatial resolution and source frequency is critical: the former affects both numerical accuracy and computational cost, while the latter determines the wavelength of the simulated wave. To satisfy the CFL condition (Smith, 1985), the time step is defined as: let $S = 0.5$ is the default Courant factor

$$\Delta t = \frac{S\Delta x}{c} = \frac{S}{c \cdot \text{resolution}}, \tag{9}$$

where $c = 1$ denotes the speed of light in Meep's normalized units. To evaluate the effectiveness of **GT-MSMW** under varying spatial and spectral conditions, we generate three datasets that simulate 2D TM polarization with different combinations of resolution and frequency:

- **2D-R0F0**: All samples are simulated with a fixed spatial resolution of 60 $pixels/\mu m$ and a source frequency of 1 $\mu m^{-1}$.

- **2D-R1F0**: The spatial resolution for each sample is randomly selected within the range [40, 80] $pixels/\mu m$, introducing variability in both spatial discretization and the corresponding time step $\Delta t$, while the frequency remains fixed at 1 $\mu m^{-1}$.

- **2D-R1F1**: Both the resolution and frequency vary. The resolution is randomly chosen from the range [40, 80] $pixels/\mu m$, and the frequency is sampled uniformly from [1, 2] $\mu m^{-1}$.

Table 2: Baseline comparisons and ablation results on four datasets.

| Model | 2D-R0F0 | | 2D-R1F0 | | 2D-R1F1 | | 3D-R0F0 | |
| | MSE $\downarrow$ | $\delta \downarrow$ | MSE $\downarrow$ | $\delta \downarrow$ | MSE $\downarrow$ | $\delta \downarrow$ | MSE $\downarrow$ | $\delta \downarrow$ |
|---|---|---|---|---|---|---|---|---|
| **GT-MSMW** | **0.0025** | **0.20**% | **0.0056** | **0.55**% | **0.0257** | **1.89**% | **0.0244** | **1.53**% |
| GAT | 0.0063 | 1.59% | 0.0410 | 4.95% | 0.0858 | 5.93% | 0.1560 | 10.84% |
| GCN | 0.0161 | 3.11% | 0.0620 | 6.52% | 0.1090 | 11.42% | 0.1276 | 9.89% |
| PINNs | 0.3530 | 60.04% | 2.1540 | 70.46% | 0.6670 | 61.03% | 1.6330 | 89.34% |
| GEO-FNO | 0.0077 | 1.61% | 0.0106 | 1.34% | 0.0353 | 2.51% | 0.0891 | 5.32% |
| DeepONet | 0.0809 | 29.81% | 0.4420 | 24.90% | 0.1810 | 21.61% | 0.2030 | 20.42% |
| MeshGraphNet | 0.0082 | 1.72 % | 0.0235 | 2.56% | 0.0489 | 2.91% | 0.3478 | 29.33% |
| EGAT-only | 0.0043 | 0.37% | 0.0371 | 2.77% | 0.0627 | 5.04% | 0.0322 | 2.96% |
| Transformer-only | 0.0470 | 5.84% | 0.1540 | 17.10% | 0.3190 | 23.10% | 0.1930 | 15.32% |

Each dataset consists of 100 samples in a $2\mu m \times 2\mu m$ square domain, enclosed by perfectly matched layers (PML). A point source polarized along the $E_z$ direction is placed inside the domain to generate wave excitation. Each sample contains a randomly positioned rectangular dielectric scatterer, which is non-magnetic, non-conductive, and non-dispersive. The scatterer's width and height are uniformly sampled from the range [0.2, 2] $\mu m$, and its relative permittivity is randomly chosen from [2, 15]. Electromagnetic field snapshots are recorded at 100 discrete time steps. For each sample, the corresponding data sequence is divided into training, validation, and test sets in an 8:1:1 ratio based on time steps. The discussion of the source term configuration and the training details are provided in Appendix B.3.

To assess the generalizability of **GT-MSMW**, we extend it to a 3D wave propagation setting within a cubic domain of size $2\,\mu m \times 2\,\mu m \times 2\,\mu m$, discretized using tetrahedral meshes. The source is configured as a plane wave in the $xy$-plane, propagating along the $z$-axis. Scatterers are generated from filtered random noise, with relative permittivity uniformly sampled from [2, 10]. We also adopt 100 scatterers in this setting, each simulated over 100 time steps, with the resolution and frequency fixed at 60 $pixels/\mu m$ and 1 $\mu m^{-1}$, respectively. This dataset is referred to as **3D-R0F0**. An 8:1:1 split over time steps is also used for training, validation, and testing. The configurations of all datasets are summarized in Tab. 1.

Notably, in the 3D setting, each vertex is associated with the electric field vector $E = (E_x, E_y, E_z)$, which represents the local electric flux, as well as the time step settings $(\Delta t, \text{step\_num})$ and the spatially varying permittivity $\epsilon(\mathbf{r})$. Similarly, each edge carries the magnetic field vector $H = (H_x, H_y, H_z)$, the time step settings $(\Delta t, \text{step\_num})$, and the relative position vector $\boldsymbol{x}_d \in \mathbb{R}^3$.

**Loss Fuction.** We use the mean squared error (MSE) as the loss function. Let $N_v = |\mathcal{V}(\mathcal{G})|$ and $N_e = |\mathcal{E}(\mathcal{G})|$ denote the number of nodes and edges in the ground truth graph $\mathcal{G}$, respectively. The loss between $\mathcal{G}$ and the prediction $\hat{\mathcal{G}}$ is defined as:

$$\text{MSE}(\mathcal{G}, \hat{\mathcal{G}}) = \frac{1}{N_v} \sum_{i=1}^{N_v} ||\boldsymbol{E}_i - \hat{\boldsymbol{E}}_i||_2^2 + \frac{1}{N_e} \sum_{i=1}^{N_e} ||\boldsymbol{H}_i - \hat{\boldsymbol{H}}_i||_2^2. \tag{10}$$

**Evaluation Metric.** The performance of various models on the four datasets is evaluated not only using MSE but also by assessing the mean relative error of the field values:

$$\delta = \frac{\sum_{i=1}^{N_v} ||\boldsymbol{E}_i - \hat{\boldsymbol{E}}_i||_2^2 + \sum_{i=1}^{N_e} ||\boldsymbol{H}_i - \hat{\boldsymbol{H}}_i||_2^2}{\sum_{i=1}^{N_v} ||\hat{\boldsymbol{E}}_i||_2^2 + \sum_{i=1}^{N_e} ||\hat{\boldsymbol{H}}_i||_2^2} \times 100\%. \tag{11}$$

## 4.2 COMPARATIVE EXPERIMENTS AND ABLATION STUDY

To provide a comprehensive benchmark, we compared our approach against several baseline models, including GCN (Kipf & Welling, 2017), GAT (Veličković et al., 2018), PINNs (Raissi et al., 2019), GEO-FNO (Li et al., 2023), DeepONet (Lu et al., 2021), and MeshGraphNet (Pfaff et al., 2021). It is worth emphasizing that while other graph-based methods such as GCN, GAT, and MeshGraphNet are able to incorporate edge features as inputs, they cannot produce edge feature outputs. Therefore, their evaluation is limited to predicting the electric field $\mathbf{E}_t$.

To quantify the respective contributions of the EGAT and Transformer modules, we conducted extensive ablation studies across all four datasets. These ablation variants are referred to as EGAT-only and Transformer-only. Details of the comparison and ablation experiments are available in Appendix B.3.

## 4.3 RESULTS

Tab. 2 summarizes the average test results of the comparative and ablation experiments across the four datasets, with MSE and relative error $\delta$ used as evaluation metrics. As indicated by the results, **GT-MSMW** consistently achieves SOTA performance across all datasets. It is particularly noteworthy that EGAT-only attains the second- or third-results, whereas Transformer-only exhibits substantially inferior performance. This observation suggests that the GNN modules contribute more substantially to the model's predictive capability than the Transformer components, which aligns with the architectural design of **GT-MSMW**—a framework primarily built upon GNNs, with Transformer blocks integrated as residual modules to enhance performance.

Fig. 3 illustrates qualitative comparisons of the predicted $E_z$ fields by **GT-MSMW** against the ground truth for three representative examples from the **2D-R1F0** test set. Each row corresponds to a distinct test instance, with columns depicting the initial condition, the ground truth solution, the **GT-MSMW** prediction, and the absolute error(AE) distribution to better highlight the differences, respectively. The high visual consistency between predictions and reference fields, together with the low-magnitude error maps, demonstrates the model's ability to capture complex field dynamics across varying initial conditions.

To further investigate the temporal evolution behavior, Fig. 4 visualizes the predicted $E_z$ fields across several time steps for the same two cases from the **2D-R1F0**. The results show that **GT-MSMW** maintains stable and accurate predictions throughout the temporal progression, confirming its effectiveness in modeling spatiotemporal behaviors Additional experimental results, including the convergence curves and relative error distributions on test sets, as well as the performance of other datasets, are presented in Appendix C.

## 5 CONCLUSIONS

This paper introduces **GT-MSMW**, a Graph Transformer framework for time-domain Maxwell's equations. For the first time, it enables direct, end-to-end prediction of electromagnetic field evolution from the initial state to arbitrary future time steps, bypassing the need for step-by-step autoregressive propagation. By integrating GNNs with residual Transformer blocks, **GT-MSMW** captures both the localized spatial interactions and the long-range dependencies critical for accurate temporal evolution. We evaluate the framework across various electromagnetic scattering scenarios, where it consistently outperforms existing approaches. Ablation studies show that the GNN modules provide strong representational capacity, while the Transformer components act as residual pathways that further enhance accuracy.

ETHICS STATEMENT

This work does not involve human subjects, personally identifiable data, or sensitive information. The datasets used in our experiments are synthetically generated by the open-source electromagnetic solver Meep, and do not raise privacy or security concerns. Our proposed methodology is intended for scientific simulation acceleration and does not produce harmful content or pose foreseeable societal risks. We have adhered to the ICLR Code of Ethics throughout this research.

REPRODUCIBILITY STATEMENT

We have taken multiple steps to ensure reproducibility of our results. Details of the proposed model architecture, training configurations, datasets, and evaluation protocols are fully described in the main paper and appendix. To further support reproducibility, we will release an anonymous zip file containing the full source code and instructions as supplementary material during the review process.

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

## A  PSEUDOCODE OF GT-MSMW FOR FDTD

Algorithm 1 presents the detailed pseudocode of the proposed GT-MSMW framework designed for multi-step field prediction in FDTD simulations.

---

**Algorithm 1:** Pseudocode of **GT-MSMW** for FDTD

---

**Input**  : Node Feature($E_0, \Delta t, \text{step\_num}, \epsilon$), Edge Feature($H_0, \Delta t, \text{step\_num}, \boldsymbol{x}_1 - \boldsymbol{x}_2$)
**Output:** $E_t, H_t$

1 **Graph Embedding:**
2 *Two independent MLP layers increase the feature dimensions of nodes and edges to 256, respectively.*
3 **for** $l \leftarrow 1$ **to** 4 **do**
4    *Four layers of **EGAT** are applied after the graph embedding, with $num\_heads = 2$ in each layer. Here, $h_i$ and $f_{ij}$ denote node and edge features, respectively.*
5    **begin**
6       Update the edge features $f'_{ij}$:
7             $f'_{ij} = \text{LeakyReLU}(A[h_i || f_{ij} || h_j])$ ;
8       Obtain attention scores $e_{ij}$:
9             $e_{ij} = \vec{F}(f'_{ij})$ ;
10       Update the node features $h'_i$:
11             $\alpha_{ij} = \text{softmax}(e_{ij})$
12             $h'_i = \sum_{j \in N_i \bigcup \{i\}} \alpha_{ij} W_t h_j$;

13 **for** $l \leftarrow 1$ **to** 2 **do**
14    *Two independent **Transformer** block layers are used to obtain $E_t$ and $H_t$, with the number of attention heads set to $n_h = 8$ and the hidden dimension to $d_h = 256$. Let $X \in R^{n \times d}$ to be the input of each Transformer layer, where $n$ is number of tokens, $d$ is the dimension of each token.*
15    **begin**
16       Compute $Q, K, V$:
17             $Q, K, V = XW_Q, XW_K, XW_V$ ;
18       RoPE Position Encoding:
19             $Q' = \text{RoPE}(Q), \quad K' = \text{RoPE}(K)$ ;
20       Multi-Head Attention:
21             $[Q'_1, Q'_2, \ldots, Q'_{n_h}] = Q'$,
22             $[K'_1, K'_2, \ldots, K'_{n_h}] = K'$,
23             $[V_1, V_2, \ldots, V_{n_h}] = V$,
24             $O_i = \text{softmax}\left(\frac{Q'_i K'^T_i}{\sqrt{d_h}}\right) V_i$ ,
25             $O = W_O[O_1, O_2, \ldots, O_{n_h}]$ ;
26       Residual connection and RMSNorm:
27             $X' = X + \text{RMSNorm}(O)$ ;
28       Feedforward network:
29             $\tilde{X} = \text{FNN}(X')$ ;
30       Final Layer:
31             $\hat{X} = \text{Linear}\big(X' + \text{RMSNorm}(\tilde{X})\big)$ .

---

# B FDTD

## B.1 COEFFICIENTS IN 2D FDTD

$$CA(m) = \frac{\frac{\epsilon(m)}{\Delta t} - \frac{\sigma(m)}{2}}{\frac{\epsilon(m)}{\Delta t} + \frac{\sigma(m)}{2}} = \frac{1 - \frac{\sigma(m)\Delta t}{2\epsilon(m)}}{1 + \frac{\sigma(m)\Delta t}{2\epsilon(m)}}, \tag{12}$$

$$CB(m) = \frac{1}{\frac{\epsilon(m)}{\Delta t} + \frac{\sigma(m)}{2}} = \frac{\frac{\Delta t}{\epsilon(m)}}{1 + \frac{\sigma(m)\Delta t}{2\epsilon(m)}}, \tag{13}$$

$$CP(m) = \frac{\frac{\mu(m)}{\Delta t} - \frac{\sigma_m(m)}{2}}{\frac{\mu(m)}{\Delta t} + \frac{\sigma_m(m)}{2}} = \frac{1 - \frac{\sigma_m(m)\Delta t}{2\mu(m)}}{1 + \frac{\sigma_m(m)\Delta t}{2\mu(m)}}, \tag{14}$$

$$CQ(m) = \frac{1}{\frac{\mu(m)}{\Delta t} + \frac{\sigma_m(m)}{2}} = \frac{\frac{\Delta t}{\mu(m)}}{1 + \frac{\sigma_m(m)\Delta t}{2\mu(m)}}. \tag{15}$$

## B.2 EXTENSION TO 3D FDTD SCENARIOS

The time-domain Maxwell's equations in 3D are given by:

$$\frac{\partial H_z(\boldsymbol{x},t)}{\partial y} - \frac{\partial H_y(\boldsymbol{x},t)}{\partial z} = \epsilon \frac{\partial E_x(\boldsymbol{x},t)}{\partial t} + \sigma E_x(\boldsymbol{x},t), \tag{16}$$

$$\frac{\partial H_x(\boldsymbol{x},t)}{\partial z} - \frac{\partial H_z(\boldsymbol{x},t)}{\partial x} = \epsilon \frac{\partial E_y(\boldsymbol{x},t)}{\partial t} + \sigma E_y(\boldsymbol{x},t), \tag{17}$$

$$\frac{\partial H_y(\boldsymbol{x},t)}{\partial x} - \frac{\partial H_x(\boldsymbol{x},t)}{\partial y} = \epsilon \frac{\partial E_z(\boldsymbol{x},t)}{\partial t} + \sigma E_z(\boldsymbol{x},t), \tag{18}$$

and

$$\frac{\partial E_z(\boldsymbol{x},t)}{\partial y} - \frac{\partial E_y(\boldsymbol{x},t)}{\partial z} = -\mu \frac{\partial H_x(\boldsymbol{x},t)}{\partial t} - \sigma_m H_x(\boldsymbol{x},t), \tag{19}$$

$$\frac{\partial E_x(\boldsymbol{x},t)}{\partial z} - \frac{\partial E_z(\boldsymbol{x},t)}{\partial x} = -\mu \frac{\partial H_y(\boldsymbol{x},t)}{\partial t} - \sigma_m H_y(\boldsymbol{x},t), \tag{20}$$

$$\frac{\partial E_y(\boldsymbol{x},t)}{\partial x} - \frac{\partial E_x(\boldsymbol{x},t)}{\partial y} = -\mu \frac{\partial H_z(\boldsymbol{x},t)}{\partial t} - \sigma_m H_z(\boldsymbol{x},t). \tag{21}$$

Similarly, by employing the Yee cell to stagger the electric and magnetic fields in time—such that their sampling is offset by half a time step—a 3D FDTD formulation can be obtained. Each magnetic field component is surrounded by four electric field components, and likewise, each electric field component is enclosed by four magnetic field components. As an example, the formulation for $E_x$ is shown below:

$$\begin{aligned} {E_x}^{n+1}\left(i+\frac{1}{2},j,k\right) &= CA(m) \cdot E_x^n\left(i+\frac{1}{2},j,k\right) \\ &+ CB(m) \cdot \left[ \frac{H_z^{n+1/2}\left(i+\frac{1}{2},j+\frac{1}{2},k\right) - H_z^{n+1/2}\left(i+\frac{1}{2},j-\frac{1}{2},k\right)}{\Delta y} \right. \\ &\left. - \frac{H_y^{n+1/2}\left(i+\frac{1}{2},j,k+\frac{1}{2}\right) - H_y^{n+1/2}\left(i+\frac{1}{2},j,k-\frac{1}{2}\right)}{\Delta z} \right] \end{aligned} \tag{22}$$

where $C_A(m)$ and $C_B(m)$ are consistent with Eqs. 12 and 13.

## B.3 IMPLEMENTATION DETAILS

**Setup of Source Term.** In our FDTD simulations, the initial condition actually corresponds to the field values across the domain after evolving for one time step $\Delta t$. At this point, we assume that one period of the source wave has already propagated through the region. Therefore, in **GT-MSMW**, we do not explicitly include the source term as an input, since its effect is already reflected in the field distribution.

To further evaluate the generalization capability of **GT-MSMW**, we construct an additional dataset, **2D-R1F1-Pulse**, by introducing a Gaussian pulse source with fwidth $= 0.2$ into the **2D-R1F1** configuration. The model is then tested on this dataset to assess its robustness under varying excitation conditions. The corresponding comparison results and temporal evolution behavior are presented in Appendix C.4.

### B.4 ILLUSTRATIONS OF 2D AND 3D MESH TRIANGULATION.

Fig. 5 demonstrates the illustrations of 2D and 3D mesh triangulation

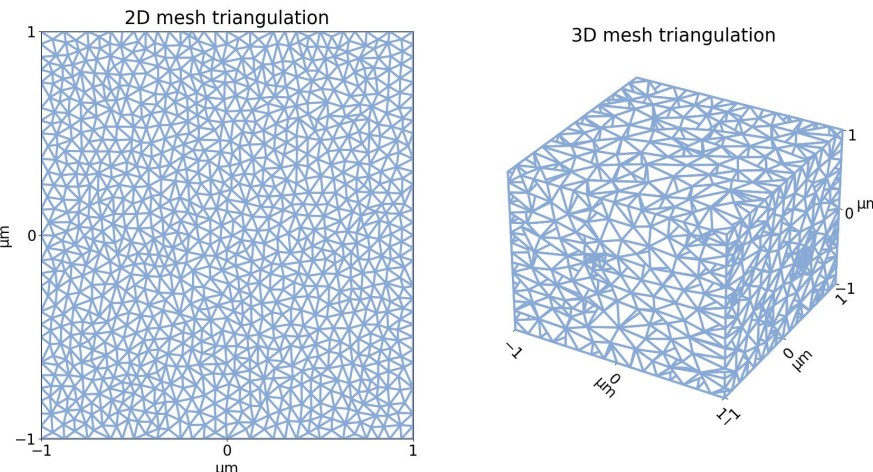

Figure 5: The left illustrates an example of 2D mesh triangulation, while the right demonstrates an example of 3D mesh triangulation.

**Training Details.** We adopt the Adam optimizer with an initial learning rate of 0.001 and beta parameters set to $[0.8, 0.999]$. A small weight decay of $3 \times 10^{-7}$ is used for regularization, and the epsilon value is set to $1 \times 10^{-8}$ to ensure numerical stability. To enhance training stability in the early stages, we employ a warm-up strategy (WarmupLR), where the learning rate increases linearly from 0 to 0.0005 over the first 100 steps. All experiments are conducted on two NVIDIA A100 GPUs with a batch size of 4 to expedite training. The average training time per epoch across the four datasets is 278s, 254s, 293s, and 131s, respectively.

**Comparative Experiments.** For the comparative experiments, the input, output, and loss function of each model are summarized in Tab. 3.

- Since GCN and GAT do not inherently support the explicit output of edge features, we restrict the update to $E_t$ in these comparative studies. Specifically, for GCN, we redesign the message-passing function to incorporate edge features into the node feature aggregation process.

- Due to the fixed-size input requirement of PINNs, GEO-FNO, and DeepONet, our evaluation is conducted on a single representative sample from each dataset. It is worth noting that, since our selected meshes are irregular, we adopt GEO-FNO as the baseline instead of FNO Li et al. (2021), which is only applicable to regular grids.

**Ablation Study.** For a fair comparison, all ablated variants preserve the same architectural scales as the full **GT-MSMW** model.

- EGAT-only: To better assess the contribution of the EGAT module, we retain the initial graph embedding layer followed by four stacked EGAT layers. Subsequently, two separate linear layers are applied to predict $E_t$ and $H_t$, respectively.

- Transformer-only: To isolate the Transformer's impact, we preserve the original graph embedding layer but replace subsequent modules with a unified two-layer Transformer architecture. The outputs are then used to predict $E_t$ and $H_t$.

Table 3: Details in comparative experiments

| Model | Input | Output | Loss function |
|---|---|---|---|
| GCN GAT MeshGraphNet | Node feature: $E_0, \Delta t, \text{step\_num}, \epsilon$ Edge feature: $H_0, \Delta t, \text{step\_num}, \boldsymbol{x}_1 - \boldsymbol{x}_2$ | $E_t$ | $\text{MSE}_{E_t}$ |
| PINNs | $(\boldsymbol{x}, t)$ | $E_t, H_t$ | $\text{MSE}_{E_t, H_t} + 0.5 \cdot \text{MSE}_0 + 0.5 \cdot \text{MSE}_f$ |
| GEO-FNO | $E_0, H_0, \Delta t, \text{step\_num}, \epsilon,$ $x_{\text{in}} = x_{\text{out}} = \boldsymbol{x}$ | $E_t, H_t$ | $\text{MSE}_{E_t, H_t}$ |
| DeepONet | Trunk net: $E_0, H_0, \Delta t, \text{step\_num}$ Branch net: $\boldsymbol{x}, \epsilon$ | $E_t, H_t$ | $\text{MSE}_{E_t, H_t}$ |

## C RESULTS

### C.1 CONVERGENCE CURVES

Fig. 6 illustrates the training process of **GT-MSMW** on the four datasets.

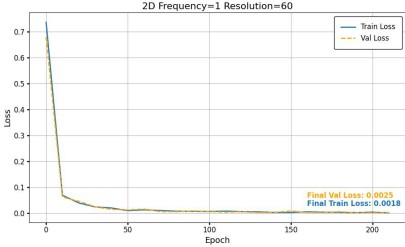

(a) The training process of **GT-MSMW** on the **2D-R0F0** dataset.

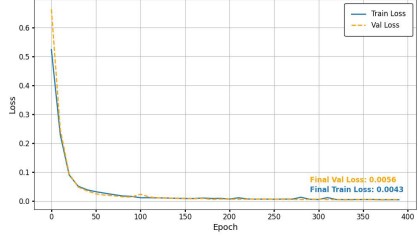

(b) The training process of **GT-MSMW** on the **2D-R1F0** dataset.

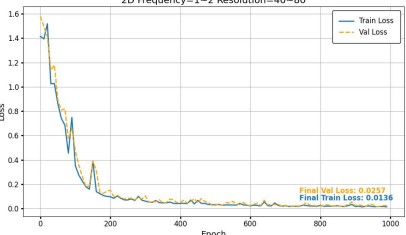

(c) The training process of **GT-MSMW** on the **2D-R1F1** dataset.

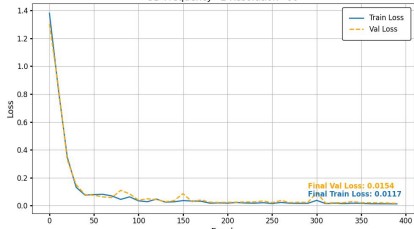

(d) The training process of **GT-MSMW** on the **3D-R0F0** dataset.

Figure 6: The training process of **GT-MSMW** on the **2D-R0F0**, **2D-R1F0**, **2D-R1F1**, and **3D-R0F0** datasets.

### C.2 RESULTS OF GT-MSMW ON THE 2D-R1F1 DATASET

Figs. 7 and 8 collectively demonstrate the temporal accuracy and robustness of **GT-MSMW** across representative test cases from the **2D-R1F1**.

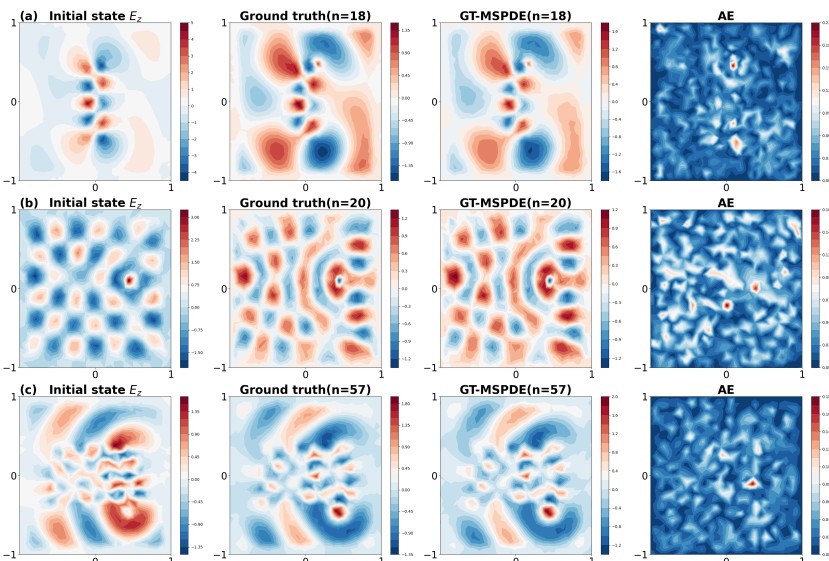

Figure 7: Visual comparisons of $E_z$ between the ground truth and the predictions from **GT-MSMW**. Subfigures (a)–(c) correspond to three representative samples from the **2D-R1F1** test set. From left to right, the columns illustrate the initial condition of $E_z$, the ground truth, the **GT-MSMW** prediction, and the associated absolute error distribution.

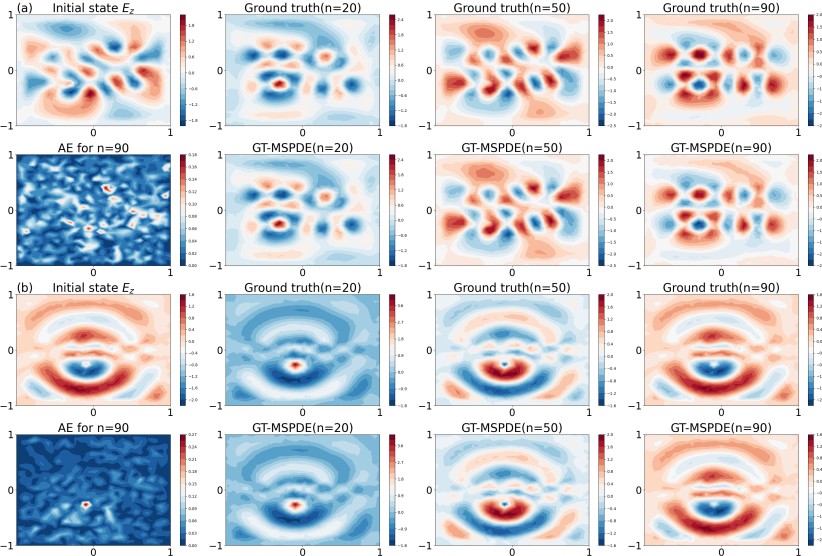

Figure 8: Visualization of $E_z$ predicted by **GT-MSMW** at different time steps for two examples from the **2D-R1F1** dataset, each with distinct initial conditions.In particular, the AE at $n = 90$ is also visualized, demonstrating that **GT-MSMW** preserves high accuracy and stability.

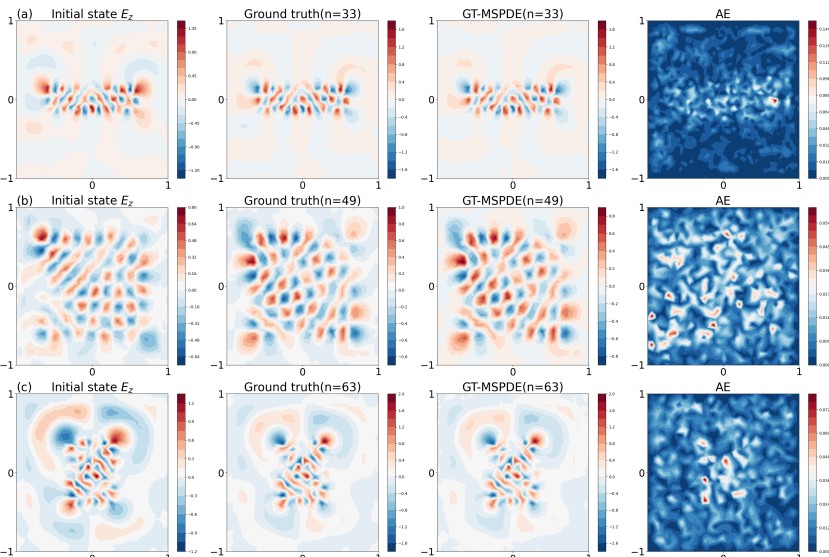

Figure 9: Visual comparisons of $E_z$ between the ground truth and the predictions from **GT-MSMW**. Subfigures (a)–(c) correspond to three representative samples from the **2D-R1F1-Pulse** test set. From left to right, the columns illustrate the initial condition of $E_z$, the ground truth, the **GT-MSMW** prediction, and the associated absolute error distribution.

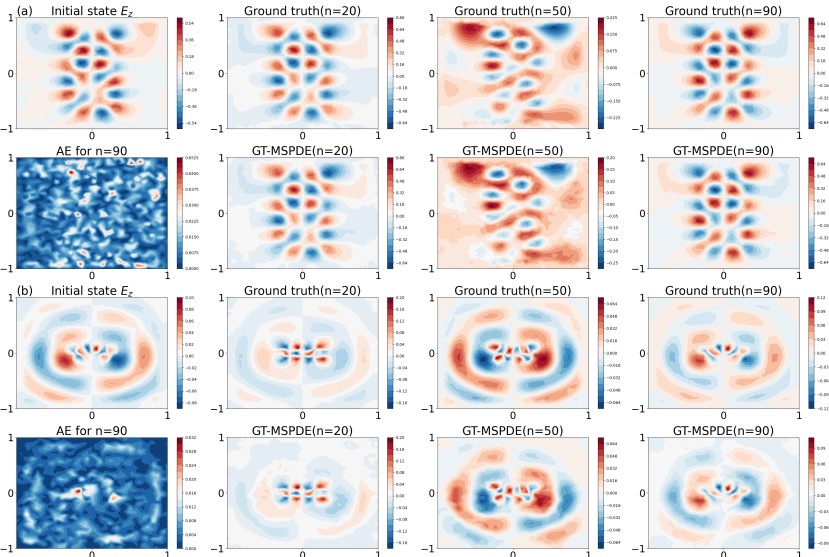

Figure 10: Visualization of $E_z$ predicted by **GT-MSMW** at different time steps for two examples from the **2D-R1F1-Pulse** dataset, each with distinct initial conditions. In particular, the AE at $n = 90$ is also visualized, demonstrating that **GT-MSMW** preserves high accuracy and stability.

## C.3   RESULTS OF GT-MSMW ON THE 2D-R1F1-PULSE DATASET

Figs. 9 and 10 provide qualitative evidence of the accurate and stable temporal performance of **GT-MSMW** from the **2D-R1F1-Pluse** set.

## D  LLM USAGE

We acknowledge the use of a large language model (LLM) solely for text polishing purposes, such as improving grammar, clarity, and readability of the manuscript. The LLM was not involved in the research ideation, experimental design, implementation, analysis, or interpretation of results. All scientific content, methodology, experiments, and conclusions are the sole responsibility of the authors.

