# OpenReview forum: "GT-MSMW: A Graph Transformer Framework for Multi-step Prediction of Time-domain Maxwell’s Equations"
_ICLR.cc/2026/Conference — ICLR 2026 Conference Withdrawn Submission_

### Official Review · Reviewer_9mJW · 2025-10-30

**Soundness:** 3
**Presentation:** 3
**Contribution:** 2
**Rating:** 2
**Confidence:** 4

**Summary:**

The paper proposes GT-MSMW, a hybrid Graph Transformer framework for multi-step prediction of time-domain Maxwell’s equations. Built upon the FDTD scheme, it uses an Edge-featured Graph Attention Network (EGAT) to model local interactions and a residual Transformer to capture long-range dependencies. The authors test the method on 2D TM polarization and 3D electromagnetic scattering problems, all within rectangular domains generated by Meep. Results show improved MSE and relative error compared with several baselines such as GCN, GAT, MeshGraphNet, GEO-FNO, and DeepONet.

**Strengths:**

- The paper is clearly written and technically consistent with standard FDTD formulations.

- It provides a reasonably complete description of the model architecture, dataset generation, and training setup.

- Combining GNNs with Transformers for spatiotemporal PDE modeling is a relevant and timely topic.

**Weaknesses:**

- Lack of motivation for GNN usage: All experiments are conducted on rectangular domains with regular grids. The claimed benefit of GNNs—flexible representation of irregular or unstructured meshes—is not demonstrated. Under the current setup, the model would behave equivalently to a CNN or Transformer, raising doubts about the necessity of GNN components. The authors should include experiments on irregular or complex geometries (e.g., curved or multi-domain interfaces) to justify why a GNN-based formulation is actually needed.



- Unclear advantage over numerical solvers: The paper does not provide any analysis or comparison against standard FDTD or other numerical solvers. There are no results regarding accuracy and computational cost, leaving it unclear why one should use this method instead of conventional solvers.

- Limited and weak experimental scope: The experiments are restricted to very simple cases. More challenging setups involving high-frequency waves, multi-interface media, or nonlinear systems (e.g., Kerr-type materials) should be included to demonstrate the method’s robustness.

- Outdated and inappropriate baselines: The baselines (GCN, GAT, vanilla PINN, GEO-FNO) are mostly outdated or not directly comparable. In particular, GEO-FNO targets operator learning on general geometries, not direct time-domain evolution. In particular, PINN has evolved substantially in recent years, with numerous improved variants addressing convergence, robustness, and accuracy. Comparing only against the plain PINN version is therefore not appropriate and fails to reflect the current state of the field. Moreover, GEO-FNO targets operator learning on general geometries, not direct time-domain evolution. Comparisons with modern PINNs and operator networks are missing.

- Lack of generalization and broader applicability: It is unclear whether the proposed method can handle nonlinear PDEs such as Navier–Stokes equations or Euler equations. The framework seems tightly coupled with the FDTD discretization, limiting its extensibility beyond Maxwell’s equations.

**Questions:**

Please refer to the detailed points listed in the Weaknesses section above.

---

### Official Review · Reviewer_LVs1 · 2025-10-31

**Soundness:** 2
**Presentation:** 3
**Contribution:** 3
**Rating:** 6
**Confidence:** 4

**Summary:**

The authors apply graph transformer architecture to design a surrogate electromagnetic solver in 2D and 3D. They check its performance for different initial values and heterogeneous materials.

**Strengths:**

-	Fairly large-scale modelling examples considered.
-	The paper is easy to follow yet technical enough, good text structuring.
-	Many baselines tested.

**Weaknesses:**

-	The authors use data from FD modelling on a square grid and then interpolate it onto a triangular grid. How exactly E- and H-fields are interpolated? Anyways, this evidently ruins the conservation properties, which are div (\eps dE/dt + \sigma E) = 0 and div (\mu dH/dt + \sigma_m H) = 0.   Consider adding illustrations on how well those properties hold for predicted fields.
-	“Magnetic conductivity” is not a standard term of general physics. Consider adding a reference to help the reader.
-	Fig. 4 shows that the absolute error reaches 5-8%. Many readers may not be happy with this accuracy. What could be done to receive more accurate predictions with the proposed architecture? Speaking more broadly, the paper misses a discussion on how prediction accuracy is controlled.
-	Fig 6. Consider plotting losses in the log scale
-	Table 2. Consider adding inference time and memory demands (at least for the 3D test case) to the table. Compare them with the respective parameters of standard FDTD modelling.
-	“The corresponding comparison results and temporal evolution behavior are presented in Appendix C.4.”. There is no such an appendix.

**Questions:**

See weaknesses

---

### Official Review · Reviewer_2Qcp · 2025-10-31

**Soundness:** 1
**Presentation:** 2
**Contribution:** 1
**Rating:** 2
**Confidence:** 4

**Summary:**

This paper introduces GT-MSMW, a hybrid Graph Transformer framework for multi-step prediction of time dependent Maxwell’s equations solution. The proposed framework integrates Edge-Featured Graph Attention Networks (EGATs) with Transformer blocks, enabling direct prediction of the electromagnetic field at an arbitrary time step which can be better than step-by-step autoregressive propagation.
The method uses the finite-difference time-domain (FDTD) inspired triangular (2D) and tetrahedral (3D) meshes. Experiments on multiple electromagnetic scattering scenarios demonstrate better accuracy compared to some alternatives.

However the paper has serious flaws and main of them is its impractical formulation. In the proposed setup in every dataset all the samples have their evolution divided to 8:1:1 train, validation and test intervals. So the solution probably won't be generalisable to unseen samples and the prediction time is only 10-20% in addition to calculated train interval, so, considering the training time, it won't be an improvement at all

**Strengths:**

* probably novel architecture combining local and global interactions
* interesting domain of study
* grids inspired by FDTD
* ablation studies
* 3D extension

**Weaknesses:**

* the impractical problem setup (no generalisation on new samples)
* competing models may be trained not optimally
* much space dedicated to all the formulas for FDTD method, however I don't feel it appropriate (it was not used in the designed model)
* the literature sometimes is cited too unselectively (too many papers in a row without differentiation between them, lines 046 or 161-162
* no stds of obtained metrics calculated

**Questions:**

* What is practical scenario of using your model? Which improvements over using FDTD scheme it will provide?
* Did you use PDE loss for training PINNs? I didn't see it in Table 3.
* What are computational expenses to train and run your model compared to numerical simulation?
* How did you use your validation dataset?
* Do you control overfitting?

---

### Official Review · Reviewer_ySz5 · 2025-11-01

**Soundness:** 2
**Presentation:** 3
**Contribution:** 1
**Rating:** 2
**Confidence:** 4

**Summary:**

The paper presents a framework that combines GNNs with transformer and FDTD for multi-step prediction of Time-domain Maxwell's Equations.

**Strengths:**

Unlike many recent works, the paper attempts to solve 3D problem.

**Weaknesses:**

Despite the fact that the paper attempts to illustrate a 3D problem, the paper is incremental. It combines FDTD with GNN and Transformer. FDTD, within Deep learning, is not new (as also cited by the author(s)).

The paper do have a strong case study and this reviewer feels it better fits as a journal article (Where the application will take the front seat) as opposed to ICLR where learning is supposed to be of primary interest.

The benchmarking should have included GraphFormer. Also, there are other graph based neural operators such as GNO, GINO, Sp2GNO. These were ignored in the paper and in the comparison section.

**Questions:**

How was the computational time during inference? I missed any mention of this. I presume inference will be expensive because of the presence of FDTD.

---

### Note · Authors · 2025-11-18

I have read and agree with the venue's withdrawal policy on behalf of myself and my co-authors.